

# Centre of pressure changes during stance but not during gait in young women after alcohol intoxication

Marta Gimunová[1], Michal Bozděch[2] and Jan Novák[2]

[1] Department of Physical Activities and Health Sciences, Faculty of Sports Studies, Masaryk University, Brno, Czech Republic

[2] Department of Physical Education and Social Sciences, Faculty of Sports Studies, Masaryk University, Brno, Czech Republic

## ABSTRACT

**Background.** Women are underrepresented in research focused on alcohol (*e.g.*, Brighton, Moxham & Traynor, 2016; DOI 10.1097/JAN.0000000000000136) despite the changing patterns of alcohol consumption, which has been increasing in women in recent decades. The purpose of this study was to analyse the relationship between habitual alcohol consumption and centre of pressure (CoP) parameters during stance and gait while intoxicated by alcohol.

**Methods.** Thirty women ($24.39 \pm 2.93$ years) participated in this study. All participants were asked to answer the AUDIT questionnaire. Stance and gait analysis were repeated under two conditions on a Zebris platform (FDM GmbH; Munich, Germany): when the participants were sober (0.00% breath alcohol concentration, BrAC) and when they were in an intoxicated state (0.11% BrAC). Participants were divided by their AUDIT score into a low-risk alcohol consumption group ($n = 15$; AUDIT score: 3 to 6) and a hazardous alcohol consumption group ($n = 15$; AUDIT score: 7 to 13).

**Results.** No statistical difference was observed in stance and gait parameters when comparing the low-risk and hazardous groups under 0.00% BrAC and 0.11% BrAC conditions. A statistically significant difference was observed when comparing 0.00% BrAC and 0.11% BrAC conditions within each group. This significant difference was found in CoP path length and CoP average velocity during quiet stance. However, no statistically significant differences were observed in CoP parameters during gait. An alcohol intoxication of 0.11% BrAC was not sufficient to cause statistically significant impairments in butterfly parameters of gait.

## INTRODUCTION

Women are underrepresented in research focused on alcohol (*Brighton, Moxham & Traynor, 2016*) despite the changing patterns of alcohol consumption, which has increased in women in recent decades. Rates of high-risk alcohol drinking have increased by 58% in women as compared to an increase of 16% in men over the past decade (*Grant et al., 2017*). In previous studies, young adults aged between 18 and 29 years were reported to have the highest prevalence of alcohol disorders and hazardous alcohol consumption

Corresponding author
Marta Gimunová,
gimunova@fsps.muni.cz

(*Turrisi et al., 2009*; *Beenstock, Adams & White, 2011*; *Davoren et al., 2016*; *Davoren et al., 2015*). The anticipation and/or direct experience of sexism may contribute to greater alcohol use among young female college students (*Petzel & Casad, 2019*). Furthermore, premenstrual symptoms (such as abdominal pain, breast tenderness, fatigue or mood lability) were observed to be a risk factor that could increase alcohol consumption among this population (*Perry et al., 2004*).

The Alcohol Use Disorders Identification Test (AUDIT) was developed by the World Health Organization (WHO). This test monitors hazardous use, dependence symptoms, and harmful use of alcohol, with scores ranging from 0 to 40 (*Lundin et al., 2015*). For women, a cut-off score of 7 was proposed to identify hazardous alcohol use, and a score above 20 indicates possible dependence (*Bradley et al., 1998*).

Acute alcohol intoxication impairs sensory and motor systems leading to a decline in postural control, which represents an increased risk of falling (*Modig et al., 2012a*; *Modig et al., 2012b*). Lower static postural stability was observed in people with alcohol use disorder (*Sullivan, Rose & Pfefferbaum, 2010*). Previous studies focusing on postural control during alcohol intoxication show statistically significant increases in anterior-posterior centre of pressure (CoP) excursions when intoxicated (*Fiorentino, 2018*; *Wu et al., 2017*). Other studies show greater medio-lateral CoP excursions when intoxicated (*Modig et al., 2012a*; *Modig et al., 2012b*; *Woollacott, 1983*).

The gait when intoxicated was characterised by decreased stride length and gait velocity (*Demura & Uchiyama, 2008*; *Hebenstreit et al., 2015*). Butterfly parameters represent the CoP trajectory during gait. In healthy subjects its diagram resembles a butterfly. Anterior-posterior variability, lateral symmetry, lateral variability, and length of gait line can be derived from the butterfly diagram. The butterfly provides information on the level of symmetry between steps, width of support, and single and double support phases (*Kalron & Frid, 2015*). Previously, changes in butterfly gait parameters have been associated with a fear of falling and balance impairments (*Kalron & Achiron, 2014*; *Kalron & Frid, 2015*).

The purpose of this study was to analyse the relationship between habitual alcohol use and CoP parameters of stance and gait in young women when intoxicated by alcohol. We hypothesised that: (i) no significant association between the AUDIT score, stance and gait parameters would be observed; (ii) greater impairments at 0.11% BrAC would be observed for stance compared to gait CoP parameters.

## MATERIALS & METHODS

Thirty women participated in this study. The inclusion criteria consisted of women between the ages of 20 and 30 years who also have an AUDIT score below 20. Exclusion criteria included pregnancy, any disease for which alcohol intake is not recommended, gait impairments, and an AUDIT score above 20. An AUDIT score above 20 indicates possible dependence. Written consent was obtained from all participants prior to data collection. The study was approved by the local ethical committee (the Research Ethics Committee of Masaryk University approved the study, EKV-2021-007) and was performed in accordance with the Helsinki declaration.

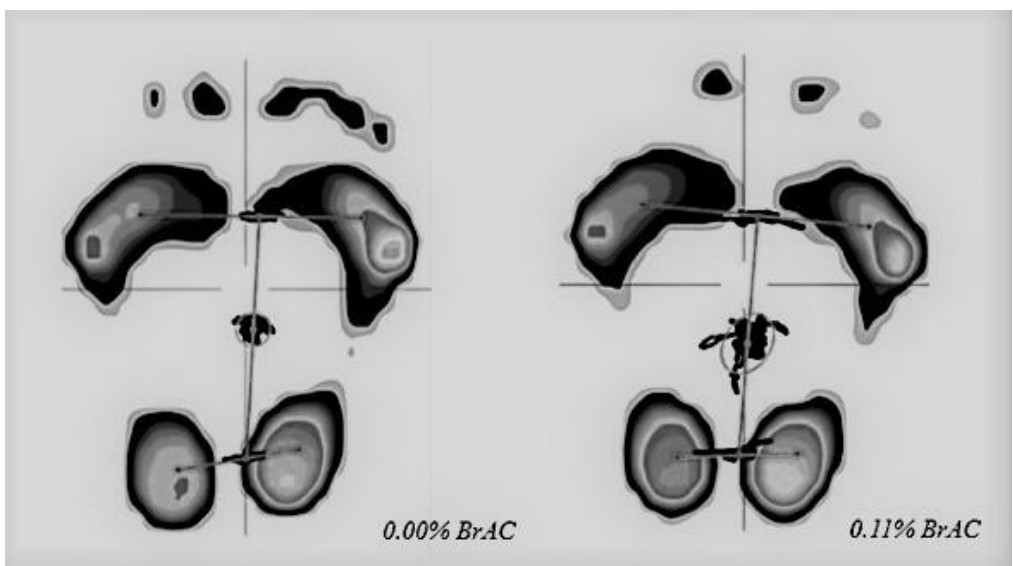

**Figure 1** **CoP path during stance analysis of one participant at 0.00% BrAC and 0.11% BrAC conditions.**

Before data collection, participants were asked to eat only a light lunch and to not consume any alcohol 24 h prior to the data collection. Data collection was performed during the afternoon hours from July to October of 2021.

All participants were asked to answer the AUDIT questionnaire. They reported their age, and their body height and body weight were measured by a stadiometer with a scale (Kern, MPE; Kern and SOHN, GmbH, Germany).

Stance analysis was performed under two conditions: when the participants were sober (0.00% BrAC, breath alcohol concentration) and at 0.11% BrAC. Participants were instructed to stand barefooted as still as possible on the Zebris platform (FDM GmbH, Munich, Germany; 149 × 54.2 cm) with a narrow stance (heels and big toes touching) for 30 s. The stability test was repeated twice under each condition (0.00% BrAC and 0.11% BrAC) and the clearer result was used for further analysis. We collected CoP path length (mm, shown in Fig. 1), CoP average velocity (mm/s), anterior-posterior CoP deviation (mm), and medio-lateral CoP deviation (mm; negative values indicate inward/medial excursion of CoP, positive values indicate outward/lateral excursion of CoP) from the Zebris software.

The procedure of gait analysis has been described previously (*Gimunová et al., 2022*). In brief, the gait analysis was performed twice; one measurement was performed after stance analysis measurement at 0.00% BrAC and the second at 0.11% BrAC. Participants were instructed to walk barefooted six times over a Zebris platform placed in a custom-designed 10-metre-long walkway surrounding the platform to provide a level walking surface. Participants were instructed to walk at their natural speed. The Zebris platform enables a detailed gait analysis of the bare foot without the perturbation caused by shoes (*e.g.*, *Klöpfer-Krämer et al., 2020*).

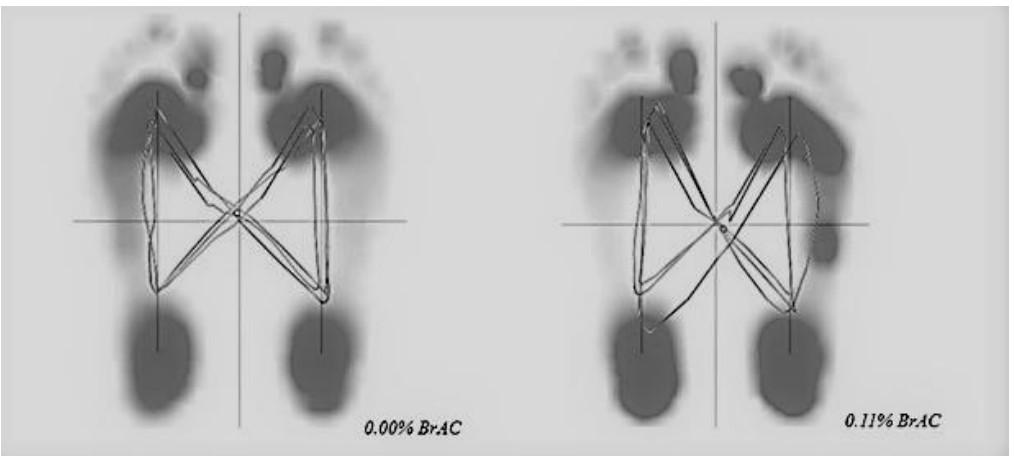

**Figure 2** **Butterfly diagrams of one participant at 0. 00% BrAC and 0.11% BrAC conditions.**

The following butterfly parameters were obtained from the Zebris software:

- Anterior-posterior position (mm): shift of the CoP intersection point in the anterior-posterior direction. Zero is the rearmost place where the heel contacts the ground.
- Anterior-posterior variability (mm): standard deviation of the CoP intersection point in the anterior-posterior direction.
- Lateral symmetry (mm): left or right shift of the CoP intersection point; zero is equivalent to perfect symmetry. A negative value indicates a shift to the left, and a positive value indicates a shift to the right.
- Lateral variability (mm): standard deviation of the lateral symmetry.
- Length of the gait line (mm): line of the CoP trajectory during the ground contact of the left or right foot.

Butterfly diagrams for 0.00% BrAC and 0.11% BrAC gait conditions are shown in Fig. 2.

Additionally, temporo-spatial gait parameters (*e.g.*, velocity {km/h}, cadence {steps/min}, stride length {cm}, and step width {cm}) were also obtained from the Zebris software.

Alcohol intoxication was performed after the first (0.00% BrAC) measurement of stance and gait. Participants consumed vodka mixed with orange juice (1:1) within a 30-minute timeframe. The amount of vodka provided to each participant was based on the body weight of each participant (1.0 g alcohol/kg body weight). After finishing the drink, BrAC was measured every 10 min with a hand-held breath alcohol testing device (Dräger Alcotest 6820; Dräger, Lübeck, Germany). The second gait measurement was performed as close as possible to 0.11% BrAC in the descending phase of alcohol intoxication. A diagram of the data collection is shown in Fig. 3.

## Statistical analysis

Most of the variables did not meet the assumptions of normal distribution, which was tested by the Shapiro–Wilk test. The Mann–Whitney U test was used to compare the differences

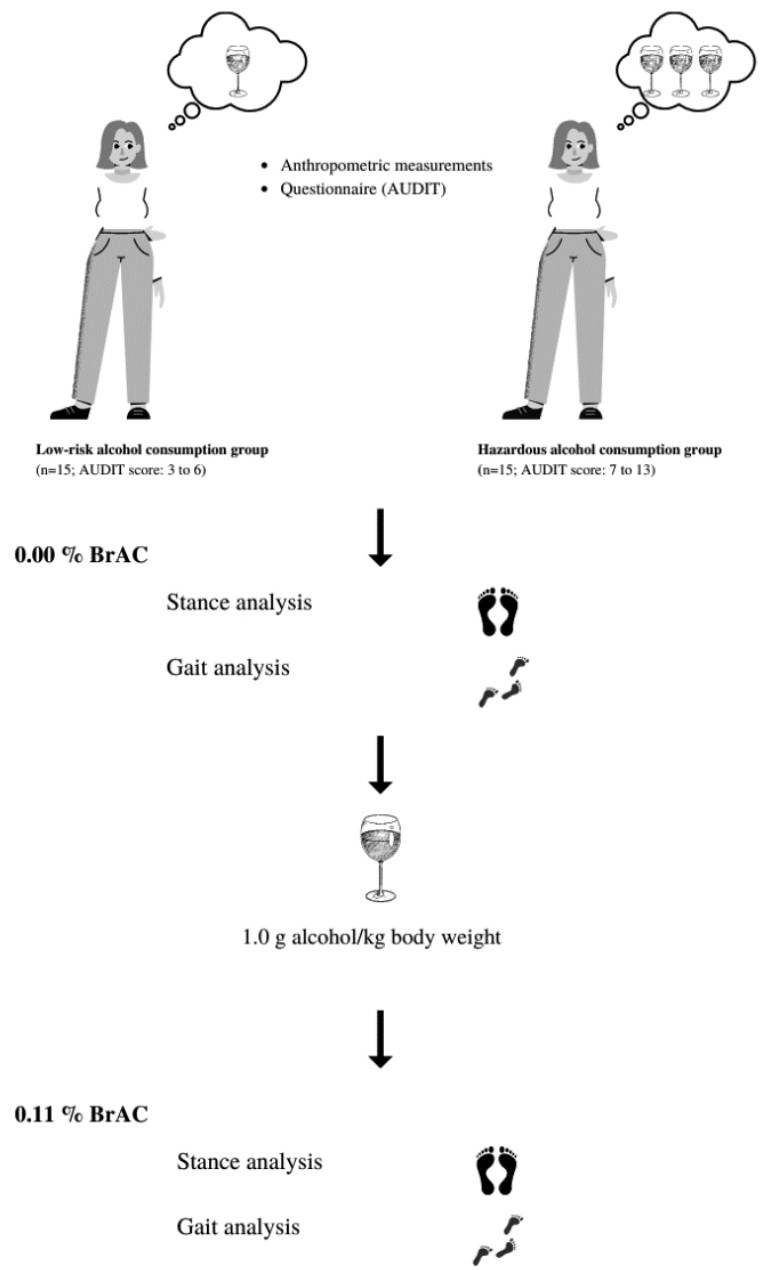

**Figure 3  Diagram of the data collection.** Canva. https://www.canva.com/.

between the characteristics of the groups, stance, and gait parameters at 0.00% BrAC and 0.11% BrAC. The Wilcoxon signed-rank test was used to compare differences between 0.00% BrAC and 0.11% BrAC conditions within each group. A Bonferroni correction was applied to an overall $\alpha$ level of 0.05 to avoid Type I error rate (adjusted $\alpha = 0.013$ for stance parameters and adjusted $\alpha = 0.005$ for gait parameters were used).

**Table 1  Low-risk consumption group and hazardous alcohol consumption group characteristics.**

|  | Low risk group ($n = 15$) Mean (SD) | Hazardous group ($n = 15$) Mean (SD) | $p$ |
|---|---|---|---|
| Body height (cm) | 168.15 (5.52) | 167.97 (5.85) | 0.740 |
| Body weight (kg) | 64.11 (7.52) | 65.72 (5.33) | 0.361 |
| Age (years) | 24.39 (2.93) | 23.74 (2.99) | 0.520 |
| AUDIT score | 4.73 (0.96) | 9.27 (1.63) | |

## RESULTS

Participants were divided by their AUDIT score into a low-risk alcohol consumption group ($n = 15$; AUDIT score: 3 to 6) and a hazardous alcohol consumption group ($n = 15$; AUDIT score: 7 to 13). No statistically significant differences were observed in age, body height or weight between the two test groups. The participants' characteristics are shown in Table 1. Mean BrAC during the second gait and stance measurement was $1.09 \pm 0.06$ for the low-risk consumption group and $1.10 \pm 0.07$ for the hazardous alcohol consumption group.

Stance CoP parameters and the results of statistical analysis are shown in Table 2. No statistical difference was observed in stance parameters when comparing the low-risk and hazardous groups under 0.00% BrAC and 0.11% BrAC conditions. A statistically significant difference was observed in CoP path length and CoP average velocity in both groups when comparing 0.00% BrAC and 0.11% BrAC within each group. Both variables were increased at 0.11% BrAC.

The spatio-temporal gait, butterfly parameters and the results of statistical analysis are shown in Table 3. No statistical differences were observed in gait parameters when comparing the low-risk and hazardous groups at 0.00% BrAC and 0.11% BrAC. A statistically significant difference in stride length was observed in the low-risk group when comparing 0.00% BrAC and 0.11% BrAC within each group. The stride length was longer at 0.11% BrAC in this group.

## DISCUSSION

The purpose of this study was to analyse changes in CoP parameters during quiet stance and gait after alcohol intoxication. To our knowledge, this is the first study into the effect of alcohol intoxication on stance and gait CoP parameters.

Participants were divided by their AUDIT questionnaire score into two groups: a low-risk alcohol consumption group (AUDIT score: 3 to 6) and a hazardous alcohol consumption group (AUDIT score: 7 to 13). Hazardous alcohol consumption is the most prevalent public health issue in young adults and university students (*Davoren et al., 2016*). Nevertheless, the majority of university health centres do not provide routine alcohol screening for students. Only 12% of students report undergoing alcohol screening using a standardised instrument (*Foote, Wilkens & Vavagiakis, 2004*). Alcohol-related consequences include

Gimunová et al. (2023), *PeerJ*, DOI 10.7717/peerj.16511

**Table 2** Stance CoP parameters at 0.00% BrAC and 0.11% BrAC conditions of low-risk consumption group and hazardous alcohol consumption group.

| | 0.00% BrAC | | | 0.11% BrAC | | | Dif. 0.00% and 0.11% BrAC | |
|---|---|---|---|---|---|---|---|---|
| | Low risk group | Hazardous group | *p* (Group dif.) | Low risk group | Hazardous group | *p* (Group dif.) | *p* (Low risk group) | *p* (Hazardous group) |
| | Mean (SD) | Mean (SD) | | Mean (SD) | Mean (SD) | | | |
| CoP path length (mm) | 280.02 (56.35) | 297.58 (89.33) | 0.663 | 510.01 (133.54) | 514.02 (243.80) | 0.443 | 0.001[*] | 0.001[*] |
| CoP average velocity (mm/s) | 9.33 (1.88) | 9.92 (2.98) | 0.663 | 17.00 (4.45) | 17.13 (8.13) | 0.443 | 0.001[*] | 0.001[*] |
| Anterior-posterior CoP deviation (mm) | 21.64 (7.53) | 16.81 (9.64) | 0.071 | 20.24 (10.21) | 19.64 (10.09) | 0.787 | 0.532 | 0.156 |
| Medio-lateral CoP deviation (mm) | −7.21 (5.98) | −4.18 (5.69) | 0.141 | −6.82 (6.45) | −4.66 (4.26) | 0.237 | 0.955 | 0.733 |

**Notes.**
*Highlights statistical significance after Bonferroni correction.

Gimunová et al. (2023), *PeerJ*, DOI 10.7717/peerj.16511

**Table 3** Spatio-temporal and butterfly gait parameters at 0.00% BrAC and 0.11% BrAC conditions of low risk consumption group and hazardous alcohol consumption group.

| | 0.00% BrAC | | | 0.11% BrAC | | | Dif. 0.00% and 0.11% BrAC | |
|---|---|---|---|---|---|---|---|---|
| | Low risk group | Hazardous group | *p* (group dif.) | Low risk group | Hazardous group | *p* (group dif.) | *p* (low risk group) | *p* (hazardous group) |
| | Mean (SD) | Mean (SD) | | Mean (SD) | Mean (SD) | | | |
| Anterior-posterior position (mm) | 6.11 (13.11) | 4.01 (5.60) | 0.917 | 3.39 (11.42) | 0.91 (6.33) | 0.373 | 0.211 | 0.088 |
| Anterior-posterior variability (mm) | 6.04 (9.37) | 2.39 (2.27) | 0.065 | 6.81 (7.95) | 3.20 (1.95) | 0.481 | 0.532 | 0.125 |
| Lateral symmetry (mm) | 1.24 (8.47) | 1.53 (5.71) | 0.576 | 2.15 (8.17) | −1.24 (3.37) | 0.351 | 0.460 | 0.054 |
| Lateral variability (mm) | 5.71 (7.99) | 2.39 (2.46) | 0.093 | 6.35 (7.95) | 3.39 (2.01) | 0.663 | 0.460 | 0.233 |
| Length of the gait line L (mm) | 205.21 (33.23) | 221.07 (13.39) | 0.254 | 214.02 (20.02) | 217.42 (19.41) | 0.820 | 0.125 | 0.910 |
| Length of the gait line R (mm) | 212.20 (21.19) | 219.03 (21.66) | 0.120 | 211.54 (25.25) | 222.75 (11.41) | 0.165 | 0.334 | 0.427 |
| Stride length (cm) | 126.53 (13.17) | 128.95 (12.06) | 0.561 | 133.30 (15.27) | 133.92 (13.84) | 0.648 | 0.004[*] | 0.009 |
| Step width (cm) | 9.63 (2.23) | 8.89 (1.84) | 0.290 | 10.13 (2.54) | 8.87 (2.53) | 0.237 | 0.307 | 0.955 |
| Cadence (steps/min) | 107.81 (11.90) | 103.95 (9.23) | 0.351 | 108.75 (12.93) | 104.98 (8.34) | 0.633 | 0.532 | 0.691 |
| Velocity (km/h) | 4.13 (0.80) | 4.04 (0.63) | 0.820 | 4.39 (0.93) | 4.22 (0.63) | 0.787 | 0.027 | 0.099 |

**Notes.**

*Highlights statistical significance after Bonferroni correction.

injuries, risky sexual behaviour (*e.g.*, unplanned and unprotected sex), decreased academic performance, driving while intoxicated, and increased psychiatric distress (*Foote, Wilkens & Vavagiakis, 2004*; *Hingson et al., 2002*; *Nolen-Hoeksema, 2004*).

Some studies suggest that women are at a higher risk of injury compared to men while in an intoxicated state (*Stockwell et al., 2002*; *Nolen-Hoeksema, 2004*). Impaired postural control was observed in alcohol-dependent patients as compared to healthy controls; this postural stability impairment was related to lifetime alcohol consumption (*Wöber et al., 1999*). The results of this study show no difference in CoP parameters between the low-risk and hazardous consumption groups during stance and gait when sober or when intoxicated.

A statistically significant difference was observed in both CoP path length and CoP average velocity when comparing stance at 0.00% BrAC and 0.11% BrAC within each group. Both variables were increased at 0.11% BrAC. A similar observation was reported in a previous study which also observed increases in CoP path length and velocity while intoxicated (*Noda et al., 2004*).

A statistically significant difference in stride length was observed in the low-risk group when comparing gait at 0.00% BrAC and 0.11% BrAC within each group. The stride length was longer at 0.11% BrAC, which could be attributed to an increase in gait velocity. However, the difference in velocity did not reach the level of statistical significance after Bonferroni correction. A previous study performed on men shows a decrease in stride length, gait velocity and cadence when intoxicated by alcohol with a blood alcohol concentration of 0.13–0.17% (*Demura & Uchiyama, 2008*). Comparison of these results suggests different motor impairments in women and men during gait when intoxicated by alcohol.

Butterfly diagrams are appropriate estimators of neurological functions, and an anterior-posterior variability was associated with cerebellar impairment in a previous study (*Kalron & Frid, 2015*). The cerebellum is vulnerable to alcohol intoxication, and alcohol-induced cerebellar ataxia leads to an increase in balance-related gait variables (*Mitoma, Manto & Shaikh, 2021*). Gait impairments were observed from 0.04% BrAC in a previous study (*Jansen, Thyssen & Brynskov, 1985*). In this study, however, alcohol intoxication of 0.11% BrAC was not enough to affect gait butterfly parameters.

There are several limitations of this study. The amount of alcohol used in this study was not sufficient to affect the CoP parameters during gait. Additionally, stance and gait measurements were performed during barefoot walking which may not be representative of daily life. Young women are used to wearing high heels or other fashionable shoes when going out to drink alcohol. Future studies should focus on women and the effects of footwear and alcohol intoxication on stance and gait parameters; this would provide greater insight into how alcohol-induced motor impairments could affect daily life as suggested previously (*Gimunová et al., 2022*).

## CONCLUSIONS

In young women, alcohol intoxication has a greater impact on CoP parameters during stance as compared to gait. The CoP variables in stance were impaired in our participants,

similarly as reported in previous studies performed on males or mixed samples. Compared to previous studies, the results of this study suggest different motor impairment in women as compared to men during gait while intoxicated by alcohol.

### Funding

This research was written at Masaryk University as part of the project number MUNI/A/1639/2020 with the support of the Specific University Research Grant provided by the Ministry of Education, Youth and Sports of the Czech Republic. The funders had no role in study design, data collection and analysis, decision to publish, or preparation of the manuscript.

### Grant Disclosures

The following grant information was disclosed by the authors:
Masaryk University:  MUNI/A/1639/2020.

### Competing Interests

The authors declare there are no competing interests.

### Author Contributions

- Marta Gimunová conceived and designed the experiments, performed the experiments, prepared figures and/or tables, authored or reviewed drafts of the article, and approved the final draft.
- Michal Bozděch analyzed the data, prepared figures and/or tables, authored or reviewed drafts of the article, and approved the final draft.
- Jan Novák performed the experiments, authored or reviewed drafts of the article, and approved the final draft.

### Human Ethics

The following information was supplied relating to ethical approvals (i.e., approving body and any reference numbers):
   Research Ethics Committee of Masaryk University approved the study (EKV-2021-007).

### Data Availability

   The raw data is available in the Supplementary File.

### Supplemental Information

Supplemental information for this article can be found online at http://dx.doi.org/10.7717/peerj.16511#supplemental-information.

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
