# Peer review of "Centre of pressure changes during stance but not during gait in young women after alcohol intoxication"

_PeerJ, doi:10.7717/peerj.16511_

## Round 0.1 · original submission · Major Revisions

Both reviewers found your manuscript to be of sufficient merit. However, both stated that the current manuscript was hard to follow due to the grammar. Please revise accordingly for the revision.

**Language Note:** The Academic Editor has identified that the English language must be improved. PeerJ can provide language editing services - please contact us at copyediting@peerj.com for pricing (be sure to provide your manuscript number and title). Alternatively, you should make your own arrangements to improve the language quality and provide details in your response letter. – PeerJ Staff

Reviewer 1 ·

Basic reporting

Overall, the authors present interesting applications of certain assessments such as their application of “butterfly parameters”. However, the authors are strongly encouraged to revise the manuscript in its entirety to improve the language used throughout the draft. In many instances, sentence structure issues were present, sentences contained errors, and require revision. Examples of this can be found throughout the abstract, and lines 39, 49, 81, 102, 111, 152, 193, 211, among others.

The authors provide relevant literature references and provide sufficient justification for the development of their research question. Furthermore, the article structure seems appropriate, with tables and figures placed in appropriate locations throughout. The authors are strongly encouraged to revise their writing in this section as it pertains to sentence structure, and grammar (such as analyse in line 69, 76, etc.), as well as ensuring consistency in the acronyms used an example being COP used in line 68 versus CoP in others. Furthermore, there requires more consistency in their reporting of variables (see line 61 and 69), there is variability in anterio-posterior vs anterior-posterior, also differs from what is used in line 117). Line 145 has 0.11BrAC, in other areas, this condition is referred to as 0.11% BrAC.

Experimental design

The authors have a well-defined research question, while also highlighting the gap they attempted to address in their efforts. The application of the butterfly parameters was novel and will facilitate applications within the general population. How their study addresses the gap is made evident by the authors. The methodology was straightforward for the purpose of replication. Methods were adequately described for ease of replication and the study appears to have met ethical standards for their institution.

Validity of the findings

Conclusions are sound and are linked to the original question. The raw data are provided. The authors present clear, well-stated conclusions without exaggerating their implications.

Additional comments

Overall, the authors appear to have conducted a well-structured study that adequately addressed the question at hand. The research appears to address a gap in the literature and does offer novel application of current approaches. However, revisions are strongly recommended for grammar, structure, and flow.

Reviewer 2 ·

Basic reporting

Thank you for taking the time to conduct/ complete the investigation examining the “Center of pressure changes during stance but not gait after alcohol intoxication in young women”. While this manuscript provides insight, it cannot be accepted in its present form.

The manuscript across the board needs to have grammar and sentence structure applied. Please send manuscript to an English expert to allow the manuscript to flow better please (broken sentence structure present throughout).

Please see below for specific comments:


Abstract:

Page 5
Line 17/18: Where is the data supporting that women are underrepresented in alcohol research?

Line 19: Analyse should be Analyze

Line 19/20/21: Delete “screened by AUDIT questionnaire” and “of stance and gait when intoxicated by alcohol in young women.” Revisit the specific examination of stance and gait while intoxicated in a following sentence.

Line 22: Specify age range instead of “young women”.

Line 24/25: Describe the participants as being in an intoxicated state and include the target BrAC in parentheses, like the “when the participants were sober” portion of the sentence for consistency.

Page 6
Line 30-31: Condition to conditions, “in both groups statistically significant difference was” to statistically significant differences were observed. End sentence there. “These significant differences were found during the quiet stance period.”

Line 31-32: Start with a however, then describe how no significant differences were found during gait.

Line 32-34: Delete entire sentence

Line 34: 0.11 into 0.11%, delete “seems not to be sufficient”. It is or it isn’t sufficient


Introduction:

Page 6:
Line 40: comma after consumption, “which is increasing” into “which has increased”

Line 41/42: Rewrite to “Over the past decade, the rates of alcohol use has increased by 16% compared to heavy alcohol consumption, which has increased by/to 58%” then cite.

Line 42/43: What’s the point? Research doesn’t focus on the underage range.

Line 44/45: delete “who are at great risk of hazardous drinking”

Line 45: delete the parentheses section regarding sexism.

Line 46: what premenstrual symptoms? Give examples in parentheses. Comma after risk factor

Line 47: Increasing the into “which could increase”, College students into “this population”

Line 48: Start sentence with “In previous studies,” Delete “college/university students and”.

Line 49: Change alcohol hazardous to hazardous alcohol. End sentence after “consumption” and the citation.

Page 7:
Line 52/53: Insert “was” after (AUDIT), Word into World, period after (WHO).

Line 53: Insert “this test” before “monitors”, insert comma after “symptoms”, insert comma after “alcohol”, delete “a”, replace “score” with “scores”.

Line 55: replace “the” with “and a”

Line 58: comma after control, falling instead of fall.

Line 59: People instead of persons, comma after disorder

Line 60: delete “some of the”

Line 61: increases instead of increase

Line 62: delete “by alcohol”

Line 63/64: delete “by alcohol”

Line 65: delete “by alcohol”

Line 67: delete the first “during the gait”, end sentence after the second “during the gait”. Begin new sentence with “in healthy subjects…”

Line 68: delete sentence “this dynamic COP…”

Line 70: comma after lateral variability,

Page 8
Line 72/73: Instead of “the butterfly gait parameters changes were…” say “changes in the butterfly gait parameters…”

Line 74/75: delete sentence.

Line 76/77: delete “screened by AUDIT questionnaire” and parentheses following
Line 79: comma after score, delete “and” before stance, end sentence after “observed”

Line 80: remove “as no participant… in this study” and instead relocate this information to exclusion criteria in methods.

Line 81: compared instead of compare

Methods & Materials:

Page 8
Line 84: rephrase “thirty young women…”,

Line 84/85: rewrite to say, “The inclusion criteria consisted of women between the ages of 20-30 years, who also have an AUDIT score below 20.” Delete the remainder of the sentence starting at “as the score above 20…”

Line 86: change when to where

Line 87: comma after recommended, delete the “and” after recommendation, comma after gait impairments, Insert information from line 80. Insert “to” after prior.

Line 92: spell out hour instead of using h, insert “to” after prior. Insert “the” after during, delete comma after hours

Line 93: use from instead of in, insert “of” after October

Line 94: end sentence at first comma, New sentence starts with “They reported…”, delete “their body”, it’s redundant, semicolon after height, use was instead of were, delete “with scale” it’s redundant.

Page 9
Line 97: “observed between” instead of “repeated at”

Line 98/99: delete “The 0.11% BrAC corresponds…” , no need for this sentence

Line 102: instead of “at” use “under”, use “clearer” instead of “better”

Line 103: “from the stance analysis…” into “we collected…”

Line 104: comma after (mm)

Line 106: delete “were obtained”

Line 110: Insert “The” before “gait”, “one measurement at 0.00% BrAC and one at 0.11% BrAC”

Line 111: comma after platform

Line 113: after participants, add “were instructed to”, and change “walked” to “walk”

Line 114: add “The” at the beginning of the sentence

Page 10:
Line 120: remove comma after positive value and add “indicates” for consistency

Lines 129-130: replace comma after parameters with a parenthesis and units with brackets; (e.g., velocity {km/h}, cadence {steps/min}, stride length {cm}, and step width {cm})

Line 132: change “during 30 minutes” to “within a 30-minute timeframe”

Line 133: omit “her” when referencing body weight; rephrase to “the amount of vodka provided was based on the body weight of each participant”

Line 134: add “a” before hand-held breath alcohol testing device

Page 11:
Line 142: add a comma and “which was” after distribution

Line 143-145: add “The” at the beginning of the sentence, add a comma after groups, omit the (low-risk consumption group and hazardous alcohol consumption group) as it is redundant at this point, add a comma after stance, and a percent sign after 0.11

Line 145-147: add “The” at the beginning of the sentence, omit “for stance and gait”, and add “A” before Bonferroni

Results:

Page 11
Line 152/153: Delete sentence, this was previously explained.

Line 154/155: “between the two test groups”

Line 155: “during” instead of “at”

Line 156: “1.09 +- 0.06 for the low-risk consumption group and 1.10 +- 0.07 for the hazardous alcohol consumption group.”

Line 164: “conditions: instead of “condition”

Page 12
Line 169: comma after spatio-temporal, delete “and”, comma after butterfly parameters

Line 171: use “differences were” instead of “difference was”

Line 172: “conditions” instead of “condition”

Line 173: percent symbol after 0.11

Discussion:

Page 12:
Line 179: “to analyse” is mentioned twice so omit one, correct “analyse” to “analyze”

Line 180: omit “and to analyse the differences between low-risk alcohol consumption group and hazardous alcohol consumption group” as it is redundant and stated many times in the sections above

Line 182: omit “performed on women” unless there are only studies looking at stance and gait CoP in men

Page 13:
Line 190: replace “when” with “while” and omit “by alcohol”

Line 193-194: rephrase the whole sentence to “Some studies suggest that women are at a higher risk of motor impairment and injury compared to men while in an intoxicated state.”

Lines 196-197: add semicolon after “alcohol-dependent patients”, add “this was” before “compared”, add “a” before “previous”, add a comma after “study, replace “and this” with “which shows the”, and omit “the” before “lifetime alcohol consumption”

Lines 198-201: omit “by alcohol”, omit lines 200 and 201, add “the two groups” after “between”, and end sentence there.
Lines 202-204: change “condition” to plural form “conditions”, omit “in both groups” and add “there was a”, omit “was”, add “both” before “CoP path length”, omit second “CoP” before “average velocity”
Lines 204-206: add “the” after “at”, add percent sign after 0.11, add “A” before “similar”, replace “the study” with “a separate study”, omit citation in the middle of the sentence, replace “who” with “which”, add “the” after “observation”, change “increase” to the plural form “increases”, replace “when” with “while”, omit “by alcohol”, add citation at the end of the sentence
Line 207: change “condition” to plural form “conditions”, add “the” before “low-risk”
Lines 208-209: add “the” before 0.11, add percent sign after 0.11, add a comma after “condition”, omit “in this group”, rephrase the last portion of the sentence to “which could be attributed to the increase in gait velocity.”
Line 210: add “the” before “Bonferroni”
Line 211: add “A” before “previous”, add “a” before “decrease”

Page 14:
Line 212: add a comma after “alcohol, replace “by” with “with a blood alcohol concentration of”, omit “blood alcohol concentration” at the end
Line 213: change “impairment” to its plural form “impairments”
Line 215: add a comma after “functions”
Line 216: add “a” before “previous”
Line 217: add “The” before “Cerebellum”, add a comma after intoxication
Line 219: add percent sign after 0.04, add “a” before “previous”
Line 224: add “a” before “barefoot”, add “phase” after “walking”, add a comma before “which”
Line 226: replace “focused” with “should focus”, change “effect” to “effects”

Lines 227-228: add a semicolon after “parameters”, rephrase the last portion of the sentence to “...; this would provide a greater insight on how alcohol-induced motor impairments could affect daily life.”
Conclusion
Line 231: omit “the” before “stance, add “as” before “compared”
Line 232: add a comma after “participants”
Page 15:
Line 234-235: omit “the”, change “when” to “while

Experimental design

See above

Validity of the findings

See above

Additional comments

See above

---

## Round 0.2 · Minor Revisions

Reviewer 2 has requested additional clarity in the methods.

Reviewer 1 ·

Basic reporting

The authors have effectively addressed the concerns from previous reviews. The article now reads well as significant modifications were made to address the concern regarding the language and sentence stucture used in the article.

Experimental design

n/a

Validity of the findings

n/a

Additional comments

The authors were effective in their revisions, all previous concerns have been addressed.

Reviewer 2 ·

Basic reporting

no comments

Experimental design

After reviewing your methods, it is unclear what type of footwear these participants wore in conducting the gait analysis ie: barefoot, tennis shoes, heels (height of heel). Shoe dimensions, etc along with the associated rationale behind the decisions made it collecting the data in this mauscript.

Validity of the findings

no comments, see above

---

## Round 0.3 · accepted · Accept

The authors have addressed the reviewers' comments. The manuscript is ready for publication.

Reviewer 2 ·

Basic reporting

n/a

Experimental design

The authors have effectively addressed the concerns from previous reviews. The article now reads well as significant modifications were made to address the concern regarding what was asked in the previous revision.

Validity of the findings

n/a

Additional comments

n/a